# Temporal Convolution Network Based Joint Optimization of Acoustic-to-Articulatory Inversion

**Guolun Sun** [1,2], **Zhihua Huang** [2,3], **Li Wang** [1,*] and **Pengyuan Zhang** [1,2]

1. Key Laboratory of Speech Acoustics and Content Understanding, Institute of Acoustics, Chinese Academy of Sciences, Beijing 100190, China; sunguolun@hccl.ioa.ac.cn (G.S.); zhangpengyuan@hccl.ioa.ac.cn (P.Z.)
2. University of Chinese Academy of Sciences, Beijing 100049, China; echohzh@163.com
3. The Xinjiang Technical Institute of Physics & Chemistry, Chinese Academy of Sciences, Urumqi 830011, China
* Correspondence: wangli@hccl.ioa.ac.cn

**Abstract:** Articulatory features are proved to be efficient in the area of speech recognition and speech synthesis. However, acquiring articulatory features has always been a difficult research hotspot. A lightweight and accurate articulatory model is of significant meaning. In this study, we propose a novel temporal convolution network-based acoustic-to-articulatory inversion system. The acoustic feature is converted into a high-dimensional hidden space feature map through temporal convolution with frame-level feature correlations taken into account. Meanwhile, we construct a two-part target function combining prediction's Root Mean Square Error (RMSE) and the sequences' Pearson Correlation Coefficient (PCC) to jointly optimize the performance of the specific inversion model from both aspects. We also further conducted an analysis on the impact of the weight between the two parts on the final performance of the inversion model. Extensive experiments have shown that our, temporal convolution networks (TCN) model outperformed the Bi-derectional Long Short Term Memory model by 1.18 mm in RMSE and 0.845 in PCC with $\frac{1}{4}$ model parameters when optimizing evenly with RMSE and PCC aspects.

**Keywords:** acoustic-to-articulatory inversion; temporal convolution network; Mean Square Error; Pearson Correlation Coefficient

## 1. Introduction

Acoustic-to-articulatory inversion (AAI), also known as speech inversion, has been extensively researched in the past years. AAI is a process to find the accurate mapping from the acoustic signal to specific articulatory features, usually concerned with human speech production and articulator movements. The process can be deemed as an ill-posed non-linear regression problem [1] since the ubiquitous one-to-many mapping from speech signals to different combinations of articulators' movements exists. Albeit, the accurate reconstruction of articulatory features is proved to be efficient sub information in several domains, including Automatic Speech Recognition (ASR) [2–5], Text-to-Speech synthesis [6,7], speech accent conversion [8], pathological speech detection [9,10], and Computer-Aided Language Learning (CALL) [11–13].

Real-time articulatory data is obtained mainly through techniques such as Electromagnetic Articulography (EMA) [14], X-ray microbeam [15], or real-time Magnetic Resonance Imaging(rtMRI) [16]. These methods require highly expensive devices and are time-consuming. In order to reduce the cost, it's of great importance to develop a method to convert the acoustic signals into articulatory trajectories with the help of statistic models when there is need to use these features. Another problem is that the articulatory trajectories at the very moment are influenced by both present and hstorical acoustic features, which requires a time-series regression model.

In previous studies, many statistic methods are conducted for the lack of data, including the codebook-based approach [17], which is to construct a codebook mapping the

acoustic feature to corresponding articulatory patterns. Kalman filtering [18] and Gaussian Mixture Model (GMM) [19] approaches are also used to model the inversion process. Hidden Markov Model is used in [20] to cope with time-series signal modeling. The two synchronous time series of acoustic and articulatory data are trained separately for each speaker. The HMM state of the acoustic model is then mapped to the articulatory state when inversion.

Due to the rapid progress in machine learning techniques, especially the breakthrough of Deep Neural Network (DNN), and the availability of several open-source articulography datasets, several traditional statistic inversion methods are updated by various machine learning architecture. DNN architectures are implemented in [21,22] and deep Mixture Density Networks (MDN) in [23]. Deep Recurrent Neural Network (RNN) brings the capability to model the time-series contextual information. Bi-LSTM, a bidirectional RNN using memory units, can achieve better performance than DNN AAI system [24]. However, Bi-LSTM models often have problems in their large amount of parameters and the difficulty to train. Furthermore, an RNN architecture processes the time-series sequentially, which brings difficulty in parallelism in the inferring stage. For the reasons above, we are in great need to consider an alternative to the Bi-LSTM models.

Convolution Neural Networks have shown remarkable potential in sequence modeling and predicting. In this paper, we proposed a novel Temporal Convolution Network-based AAI system, inspired by [25]. Our system uses convolutional layers to learn the contextual information of the time-series acoustic features. A joint-optimizing target function is arranged, combining two common metrics, RMSE and PCC together. The model will be optimized from the absolute differences aspect and the overall trend jointly using this target function with the help of the multitask training method The proposed system performs better than the Bi-LSTM method in the two aspects with a smaller model size. The methodology and detailed results are discussed in Sections 2 and 5.

## 2. Methods

In this section, we mainly discuss the detailed implementation of a TCN block in Sections 2.1 and 2.2, the joint optimization method.

### 2.1. Temporal Convolution Network

The Temporal Convolution Network (TCN) was first introduced in [25]. This architecture uses dilated convolution layers to transform the input sequence information to the synchronous output sequence. When dealing with the speech signals, we first get the input speech spectrum $\mathbf{X} : \{x_0, x_1, \cdots, x_t\}$ as a t-frame sequence where $x_t \in \mathbb{R}^f$ represents the feature vector at the $t$th frame. And we wish to get an output $\mathbf{X}' : \{x'_0, x'_1, \cdots, x'_t\}$ as the frame synchronous $c$-dimension embedding $x'_t \in \mathbb{R}^c$ for every single frame. The sequence model network is any function $f : X \Rightarrow X'$ that produces the mapping as Equation (1).

$$x'_0, x'_1, \cdots, x'_t = f(x_0, x_1, \cdots, x_t) \tag{1}$$

A TCN block is comprised of dilated convolution layers and a residual connection. The dilated convolution layers are the key factors in a TCN block to produce the same length output sequence as the input, using the sequence's contextual information. Figure 1 illustrates an example for a 3-stack dilated convolution layers with a kernel size as 3 and the dilation as 2. The kernel size determines the size of the convolution, and the dilation indicates refers to the number of intervals between the points of the convolution kernel. These two parameters, alongside the layer depth n, determines the reception field of the sequence model. The larger the reception field is, the more contextual information can be seen when conducting the convolution at the time-step t. The reception field is computed as Equation (2):

$$\omega = 1 + (k-1) \times \frac{d^n - 1}{d - 1} \tag{2}$$

where $\omega$ is the reception field, $k$ is the kernel size, $d$ is the dilation base and n represents the layer number.

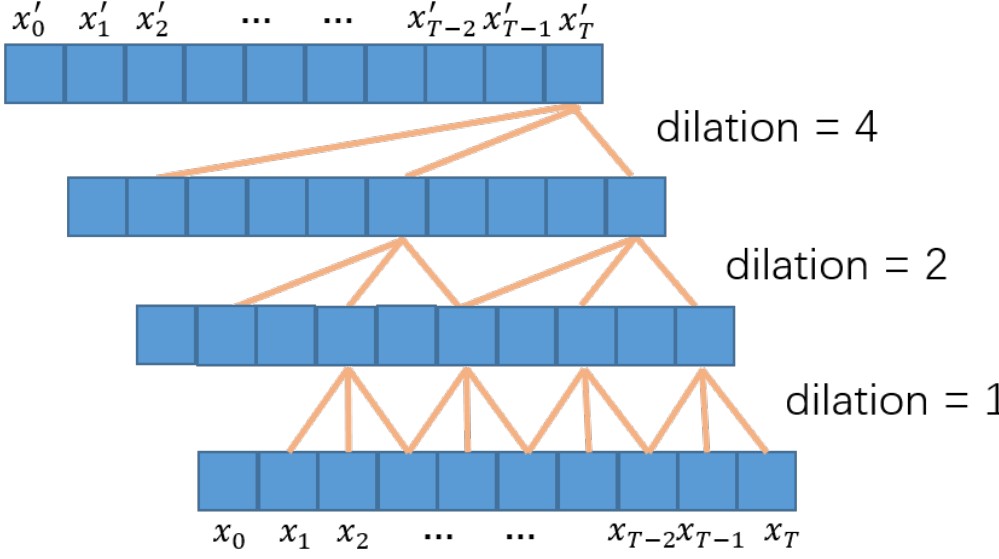

**Figure 1.** The visualization of a stack of dilated convolution layers which transform the input sequence **x** into **x'** with kernel size = 3. The dilation_base is set to be 2. 3 stacks of dilated conv-layers acquire a reception field of 15 frames.

Since the reception field is positively correlated with the depth of the convolution layers, we need to train a deeper model. The residual connect is proposed so that a relatively deep network can be trained [26]. A residual connect contains an additional output branch leading to the model $\mathcal{F}'s$ final output unit, converting the output $o$ to:

$$o = Activation(x + \mathcal{F}(x)) \tag{3}$$

As shown in Figure 2, we stack two dilated convolution layers with Rectifier Linear Unit (ReLU) [27] as the activation function. Weight normalization and dropout techniques are introduced into the construction of our final TCN block in order to avoid over-fitting.

In addition to the TCN layer, the entire model consists of two linear layers to first transform the acoustic feature $\mathbf{X} \in \mathbb{R}^{t \times f}$ into hidden space with a feature sequence $\mathbf{X}^D \in \mathbb{R}^{t \times h}$, a feed-forward readout layer to get the final 16-dim articulatory trajectories, and finally a post-filter layer for smoothing the output sequences. The Figure 3 illustrate the stacks of the model architecture.

The post-filter is implemented with reference to [28] by employing a $1 \times 1$ fixed-param convolutional layer. A 5-order low-pass filter is used to smooth the output trajectories in the conv1d layer. This can avoid dispensable computation of error back-propagation due to the jitter of the unsmoothed predicted trajectories during the training stage. The target EMA sensor recordings are smoothed by the same filter at the data preprocessing stage as well to avoid target mismatch.

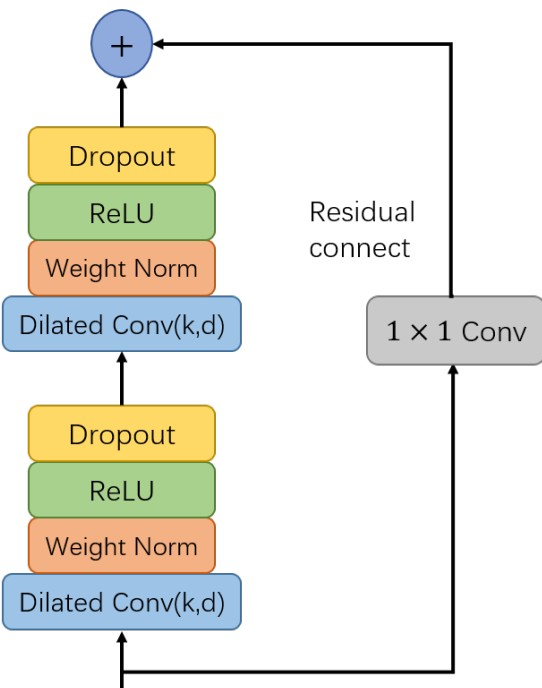

**Figure 2.** A simple implementation of the TCN residual block using in the experiment. k represents the kernel size and d represents the dilation base.

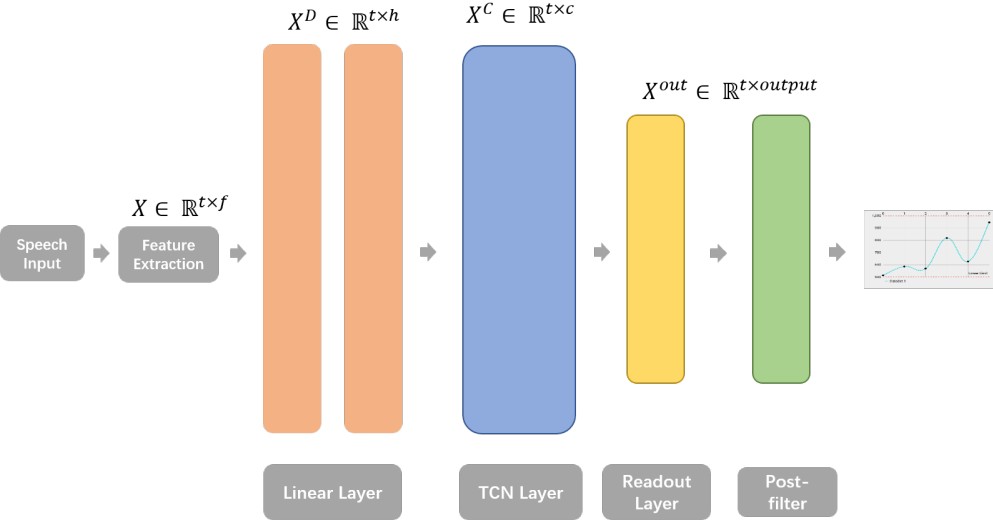

**Figure 3.** The visualization of the TCN-based AAI model.

### 2.2. Joint Optimization Target Fuction

Two conventional measurements are used in previous studies [21,22,28,29], Root Mean Square Error (RMSE) and Pearson Correlation Coefficient (PCC), which are computed as Equations (4) and (5). RMSE is used to describe the deviation between predicted sequences and targets, and the PCC mainly shows the similarity of overall trend.

Mark $y'(i)$ as the $i$-th predicted articulatory trajectories and $\bar{y}$ and $\bar{y}'$ as the average of the sequence $y$ and $y'$.

$$RMSE(y',y) = \sqrt{\frac{1}{n}\sum_{i=1}^{N}(y'(i)^2 - y(i)^2)} \tag{4}$$

$$PCC(y',y) = \frac{\sum_{i=1}^{N}(y'(i) - \bar{y}')(y(i) - \bar{y})}{\sqrt{\sum_{i=1}^{N}(y'(i) - \bar{y}')^2 \sum_{i=1}^{N}(y(i) - \bar{y})^2}} \tag{5}$$

Note that the two measurements are conventionally used separately when judging the model performance. The regression model, however, is trained with the frequently-used RMSE loss function. That leads to a mismatch of targets in the evaluation and training stage that it will care more to ease the definite error but not the sequence trend. In order to eliminate the mismatch during the training stage and optimize the performance from the two aspects aforementioned, we combine these two measurements as the model target function $\mathcal{L}(x)$ as Equation (6) for model error back-propagation:

$$\mathcal{L}(x) = \lambda RMSE(f_{INV}(x), y) - (1 - \lambda)PCC(f_{INV}(x), y) \tag{6}$$

The $\lambda \in [0, 1]$ is the weight parameter adjusting the ratio of the two parts target function. And $f_{INV}$ is the inversion model function. $x$ is the input acoustic feature. The value of the weight parameter $\lambda$ can be set in the model initialization process. Both RMSE and PCC have been appropriately scaled to ensure that the two parts of loss are under the same measure.

## 3. Dataset

### 3.1. Dataset Description

Electromagnetic articulograph (EMA) is one of the promising techniques to acquire acoustic-articulatory data. Two open-source EMA data sets, USC-TIMIT [30] and IEEE-EMA [14] corpus, which contain acoustic data in English along with EMA data are used in our AAI experiment. USC-TIMIT corpus provides EMA data for four native speakers (2 males and 2 females) of General American English, each person with 460 utterances. IEEE-EMA corpus consists of eight speakers' (4 males and 4 females) 720 utterances and EMA trajectories. The articulatory data of both datasets were collected using Northern Digital Inc. System. Thus, the articulatory features share the same dimensions and correspond to the same organs.

### 3.2. Articulatory Features

Sensors in the midsagittal plane are used in order to record the movement information of specific organs including tongue blade (TB), tongue tip (TT), tongue dorsum (TD), upper lip (UL), lower lip (LL) and lower incisor (LI). The recorded data are then measured in the sagittal plane with $x$, $y$ dimensions and then we get 12 articulatory trajectories. Figure 4 shows the articulators in the midsagittal plane.

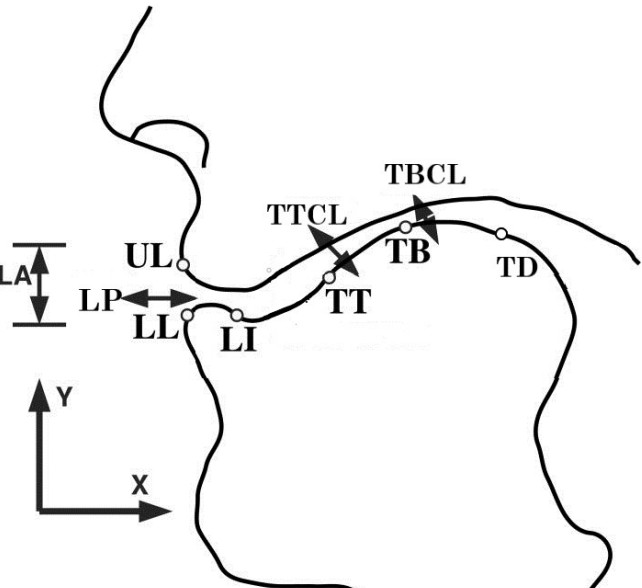

**Figure 4.** A simple schematic diagram showing the concerning articulators.

*3.3. Vocal Tract Variables*

In addition to the conventional EMA sensor recording, we compute other four dimension vocal tract variables (TVs) to describe the articulators' movements more explicitly, as is mentioned in previous work [28,29]. Some slight modifications are conducted with Equations (7) and (8):

$$TTCL = \frac{TT_x}{\sqrt{TT_x^2 + TT_y^2}} \tag{7}$$

$$TBCL = \frac{TB_x}{\sqrt{TB_x^2 + TB_y^2}} \tag{8}$$

TTCL (tongue tip cosine location) and TBCL (tongue blade cosine location) represent the cosine of the angle that the speaker's tongue tip and tongue blade formed in the *x-y* plane. Besides, LA (lip aperture) and LP (lip protrusion) are computed as Equations (9) and (10)

$$LA = \sqrt{(UL_x^2 - LL_x^2) + (UL_y^2 - LL_y^2)} \tag{9}$$

$$LP = \frac{UL_x + LL_x}{2} \tag{10}$$

The LA feature represents the Euclidean Distance between lower lips and upper lips. The LP feature represents the average of the horizontal coordinates of lips. In total, there are 16 dimensions of articulatory features for prediction.

## 4. Experimental Setup

*4.1. Feature Extraction*

The audio file is resampled to 16 kHz. We employed the Mel-Frequency Cepstral Coefficients (MFCC) as the acoustic features in the AAI experiments. We used a Hamming window with 25 ms frame length and 10 ms frameshift in the feature extraction stage, and extracted 13 cepstral coefficients per frame, where delta and 2-delta are computed. We also conducted the Cepstral Mean and Variance Normalization (CMVN) and Vocal Tract Length Normalization (VTLN) to normalize the feature and to reduce the differences among speakers [29].

For the articulatory trajectories, we did a smoothing process with the low-pass filter mentioned above and resampled to 100 Hz sequences, assuring the synchronization of the acoustic feature and articulatory trajectories. The additional 4 dimension articulatory trajectories are computed after the pre-processing stage.

*4.2. Model Parameters*

In the experiment, we mainly conduct the experiments on the IEEE-EMA corpus. The USC-TIMIT corpus is used to test the cross corpus performance. We divided the corpus into 7:2:1 for train, valid and test stage separately. Specifically, we choose speakers M04 from IEEE-EMA corpus and F5 from USC-TIMIT corpus as valid data and speaker M01 from IEEE-EMA as test data.

For the Bi-LSTM baseline system, we adopted the 2-stacks Bi-LSTM in [24,28] with 512 hidden units to replace the TCN layer mentioned in Figure 3. We keep other system architecture to be exactly the same. The TCN block is a 6-layer dilated convolution stack with different convolution channels. We set the kernel size to be 3 and dilation base to be 2, which extend the reception field to 127 frames. The first two feed-forward dense layers was initialed with 256 hidden units. It convert the 39-dim MFCC into a higher feature space. The read-out layer and the convolutional post-filter layer is set to have the same output dimension with the overall predicted trajectories. The model output is frame-synchronized.

The Adam optimizer and early stopping strategy were employed, and the initial learning rate was 0.001. L2 normalization was adopted in the experiments with a weight decay of 0.001. The entire implementation was conducted using PyTorch Toolkit.

## 5. Results

### 5.1. TCN Model vs. Bi-LSTM

First, we compare the performance of the proposed TCN model with the conventional Bi-LSTM. The Bi-LSTM model follows the parameters aforementioned in Section 4. The TCN net was implemented with different channel numbers to evaluate the impact of the number of parameters. The target function weight $\lambda$ was set to be 0.5 to optimize the model from 2 aspects evenly. Table 1 demonstrates the overall performance of the models in RMSE, PCC, and number of the parameters (Params) 3 aspects. All the 16-dim articulatory features are taken into account and the RMSE and PCC are computed per output dimension and then average.

**Table 1.** Comparison between proposed TCN-based AAI model with different channel number and Bi-LSTM baseline model. The TCN model outperformed the baseline with much fewer parameters.

|  | RMSE (SD) | PCC | Params |
| --- | --- | --- | --- |
| Bi-LSTM | 1.44 (0.46) mm | 0.770 | 11.0 M |
| TCN-512channel | 1.09 (0.32) mm | 0.873 | 9.9 M |
| TCN-256channel | 1.18 (0.40) mm | 0.845 | 2.5 M |
| TCN-128channel | 1.30 (0.46) mm | 0.801 | 0.6 M |

Bi-LSTM baseline model achieves 1.44 mm mean RMSE with 0.46 standard deviation and 0.77 Pearson Correlation Coefficient, which is consistent with [28]. When the channel number is set to be 512, the TCN model greatly outperforms the baseline with relatively comparable params. The mean RMSE decreases from 1.44 mm to 1.09 mm (24.3%) with 0.32 standard deviation and PCC improves to 0.873 (13.4%). However, this brings a lot more computation during training and evaluation stages, and it's fairly hard to train. 256-channel TCN model gets 1.18 mm RMSE (18.1% decrease) with 0.43 standard deviation and 0.845 PCC (9.7% increase). Note that the params is only $\frac{1}{4}$ of Bi-LSTM. The performance of the 128-channel model slightly precedes the baseline with a nearly $\frac{1}{20}$ model size. The experiment results depict that a deep temporal convolution network can outperform the RNN architecture with a smaller model size at the meantime when modeling the time sequence contextual information.

Figure 5 demonstrates predicted trajectories of the tongue tip articulator for a single speech recording randomly chosen from the test set.

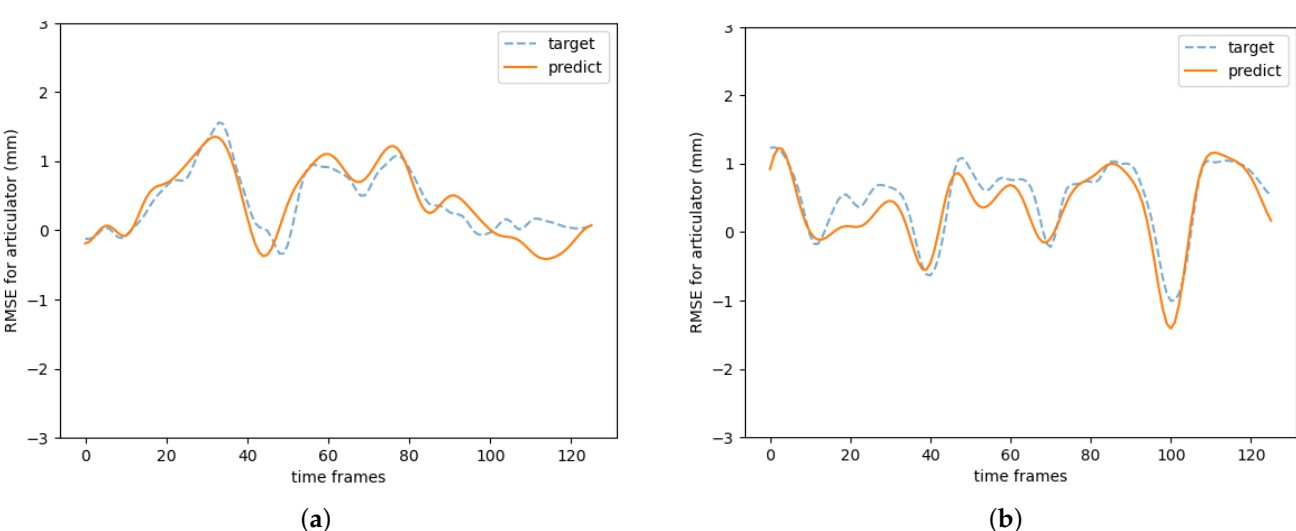

**Figure 5.** Trajectories of one specific articulator: tongue tip predicted by the proposed 256-channel TCN AAI system. The plot of the movements shows the target trajectory and the predicted one. (**a**) TT_X; (**b**) TT_Y.

### 5.2. Weight Analysis of Target Function

Another analysis on the influence of the target function weight $\lambda$ is conducted, in which we fix the TCN model parameters to 256 channels, and change the $\lambda$ from 0 to 1 with a stride of 0.1. 11 sets of contrast experiments are recorded. The Table 2 shows the results in the contrast experiments.

**Table 2.** Part of the results of the contrast experiments. An evenly trained model with $\lambda = 0.5$ achieves the best performance.

| Weight $\lambda$ | RMSE | PCC | Weight $\lambda$ | RMSE | PCC |
|---|---|---|---|---|---|
| 0.1 | 1.276 | 0.812 | 0.6 | 1.200 | 0.836 |
| 0.2 | 1.238 | 0.829 | 0.7 | 1.220 | 0.820 |
| 0.3 | 1.241 | 0.825 | 0.8 | 1.285 | 0.812 |
| 0.4 | 1.219 | 0.838 | 0.9 | 1.275 | 0.812 |
| 0.5 | 1.180 | 0.845 | 1 | 1.280 | 0.811 |

The line graph results in Figure 6 demonstrate that the combination of the two-part measurement is effective in promoting the model performance in the two concerning aspects. And the model achieves the best result reported in Section 5.1 with 1.18 mm RMSE and 0.845 PCC when $\lambda$ is 0.5, which means an even optimization of RMSE and PCC is valid for improving the AAI system. We specially note that when $\lambda$ approaches 1, which means the RMSE is of predominant in the error back-propagation, the performance is analogous to that achieved with $\lambda$ approaches 0.1. Nevertheless, it can not gain any promotion from pure PCC target function as the $\lambda$ is 0, which achieves an RMSE of 15.72 mm with a 0.82 PCC. The RMSE is too large to be shown in the line graph Figure 6. This is due to the PCC only concerned with the overall trend, rather than focus on absolute error. The model will give a line with similar trajectory to the predicted target, but the absolute value, which indicates the articulators' position, achieves a rather large difference, showing that the model did not learn the mapping from acoustic features to the articulatory features.

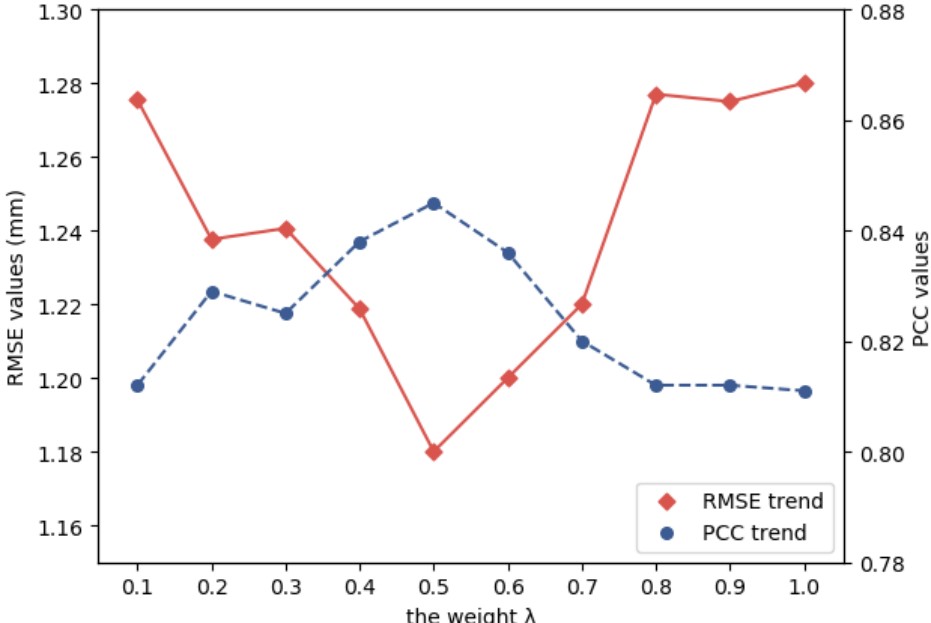

**Figure 6.** A line graph for analyze the impact of $\lambda$ on the two aspects. the red line with diamond illustrates RMSE trend, and the blue line with circle represents PCC trend. $\lambda = 0.5$ the model acquires the best performance in the two measurements' aspects. The results when $\lambda$ approaches 0.1 or 0.9 is nearly the same. But using pure PCC target achieves a fairly large RMSE which cannot be illustrated in the line graph.

## 6. Conclusions

In this study, we proposed a lightweight acoustic-to-articulatory inversion system. We built and evaluated the temporal convolution network-based AAI system with a joint optimization method on the corpus of USC-TIMIT and IEEE-EMA. We predicted a 16-dim output of the articulator movements in the midsagittal plane with the real-time sensor recordings. The performance of our system outperforms the Bi-LSTM baseline with a relative 18.1% decline in RMSE and 9.7% advance in PCC. This result shows that the novel TCN model can extract proper convolutional features for the articulatory movement reconstruction from the normalized speech features cross speakers. Changing the weight parameter $\lambda$ will affect the final performance of the AAI system. Training the model evenly on the two optimization aspects yields the best results. We leave for the future work the dynamic regulation of the target function weight and other hierarchical feature modeling methods.

**Author Contributions:** Conceptualization, G.S., Z.H., L.W. and P.Z.; methodology, G.S.; implementation G.S.; validation, G.S., Z.H., L.W. and P.Z.; formal analysis, G.S.; writing—original draft preparation, G.S.; supervision, Z.H., L.W. and P.Z. All authors have read and agreed to the published version of the manuscript.

**Funding:** This work is partially supported by the National Key Research and Development Program (No. 2020YFC2004100).

**Conflicts of Interest:** The authors declare no conflict of interest.

## Abbreviations

The following abbreviations are used in this manuscript:

| | |
|---|---|
| AAI | Acoustic-to-Articulatory Inversion |
| Bi-LSTM | Bi-directional Long Short Term Memory |
| TCN | Temporal Convolution Network |
| RMSE | Root Mean Square Error |
| PCC | Pearson Correlation Coefficient |
| ASR | Automatic Speech Recognition |
| CALL | Computer-Aided Language Learning |
| EMA | Electro-Magnetic Articulography |
| rtMRI | real-time Magnetic Resonance Imaging |
| GMM | Gaussian Mixture Model |
| HMM | Hidden Markov Model |
| DNN | Deep Neural Networks |
| MDN | Mixture Density Networks |
| RNN | Recurrent Neural Networks |
| CNN | Convolution Neural Networks |
| MFCC | Mel-Frequency Cepstral Coefficients |

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
