# Peer review of "Temporal Convolution Network Based Joint Optimization of Acoustic-to-Articulatory Inversion"

_applsci, doi:10.3390/app11199056_

Round 1

Reviewer 1 Report

The paper describes a work on the usage of "Temporal Convolution Network" to improve the development of speech-articulatory inversion mapping system. Further, a multi-task learning loss function is also proposed, which is based on the objective measurements for the accuracy of articulatory trajectory. The results have shown the effectiveness of the proposed system.

My comments are as follows:

There is a phrase of "Joint Optimization" on the title. One would expect that the paper describes a speech-articulatory inversion system that is jointly trained with another system with different objective, e.g., articulatory-to-speech production system. However, it turns out that the aforementioned phrase refers to only the usage of a multi-task learning, namely with RMSE and PCC measures, which is only mentioned briefly in the last paragraph of Section 1. Please carefully describe this point to avoid confusion for readers.

There exist unsuitable phrasing and sentence constructions throughout the manuscript. Let me give a one short example from Section 1 Paragraph 2:
    - "Electroralmagnetic ..."
        --> Electromagnetic
    - "Thus, it's of great importance to convert the acoustic signal into articulatory features."
        --> One would interpret it as if an acoustic signal is not converted into articulatory features, then an acoustic signal cannot be utilized properly.
    - "Another problem is that the articulatory trajectories at the very moment are concerned with not only the current acoustic features but also the past states, ..."
        --> In addition of implying that articulatory trajectories have some kind of consciousness, this phrase also implies that past articulatory states are treated as objectives to be optimized; while in practice, they are utilized to assist the system development for estimating the current articulatory state.

Furthermore, please do not use word "bad" or "badly" in a scientific document. Could the authors please revise the manuscript with proofreading?

In addition, it would be helpful if the authors also show the performance when setting the loss function in Eq. (4) without weighting coefficient, i.e., simply sum the two measures as they are. It is likely that it would exhibit the same tendency with 0.5 coefficient, but with differences in the time to reach the model convergence. If investigated properly, it may have a benefit that such treatment could avoid overfitting on training dataset.

Reviewer 2 Report

The article presents a novel and interesting method of speech inversion. Besides my minor remarks, I think that section 2.1 should be more substantially improved. Presently is not "reader-friendly". My minor remarks are given below and the rest are in the comments in the attached PDF.

1) TCN, Bi-LSTM - full names should be given during the first use of the abbreviations (independently from the dictionary of abbreviations at the end of the article). 

2) It seems that the article "Hybrid convolutional neural networks for articulatory and acoustic information based speech recognition" (DOI: 10.1016/j.specom.2017.03.003) could be included in the references.

3) line 73-74 "... be trained. [25]" -> "... be trained [25]."

4) What is "Params" in table 1? The explanation should be given in the text.
